# Does IPV Boost Intestinal Immunity among Children under Five Years of Age? An Experience from Pakistan

**DOI:** 10.3390/vaccines11091444

**Published:** 2023-09-01

**Authors:** Muhammad Atif Habib, Sajid Bashir Soofi, Imtiaz Hussain, Imran Ahmed, Zamir Hussain, Rehman Tahir, Saeed Anwar, Simon Cousens, Zulfiqar A. Bhutta

**Affiliations:** 1Centre of Excellence in Women and Child Health, Aga Khan University, Karachi 74800, Pakistansajid.soofi@aku.edu (S.B.S.);; 2Department of Pediatrics & Child Health, Aga Khan University, Karachi 74800, Pakistan; 3Trust for Vaccines and Immunization, Karachi 74400, Pakistan; 4Prime Institute of Public Health, Peshawar 25160, Pakistan; 5London School of Hygiene and Tropical Medicine, London WC1E 7HT, UK; 6Centre for Global Child Health, The Hospital for Sick Children, Toronto, ON M5G 1X8, Canada

**Keywords:** Polio, IPV, OPV, intestinal immunity, humeral immunity, Pakistan

## Abstract

The oral poliovirus vaccine (OPV) has been the mainstay of polio eradication, especially in low-income countries, and its use has eliminated wild poliovirus type 2. However, the inactivated poliovirus vaccine (IPV) is safer than OPV, as IPV protects against paralytic poliomyelitis without producing adverse reactions. The present study compared mucosal and humoral responses to poliovirus vaccines administered to previously OPV-immunized children to assess the immunity gap in children in areas of high poliovirus transmission. A cluster-randomized trial was implemented in three high-risk districts of Pakistan—Karachi, Kashmore, and Bajaur—from June 2013 to May 2014. This trial was community-oriented and included three arms, focusing on healthy children below five years of age. The study involved the randomization of 387 clusters, of which 360 were included in the final analysis. The control arm (A) received the routine polio program bivalent poliovirus vaccine (bOPV). The second arm (B) received additional interventions, including health camps providing routine vaccinations and preventive maternal and child health services. In addition to the interventions in arm B, the third arm (C) was also provided with IPV. Blood and stool samples were gathered from children to evaluate humoral and intestinal immunity. The highest levels of poliovirus type 1 serum antibodies were observed in Group C (IPV + OPV). The titers for poliovirus type 2 (P2) and poliovirus type 3 (P3) were noticeably higher in those who had received a routine OPV dose than in those who had not across all study groups and visits. Providing an IPV booster after at least two OPV doses could potentially fill immunity gaps in regions where OPV does not show high efficacy. However, IPV only marginally enhances humoral immunity and fails to offer intestinal immunity, which is critical to stop the infection and spread of live poliovirus in populations that have not been exposed before.

## 1. Introduction

The Global Polio Eradication Initiative (GPEI) was established in 1988, and since then, it has reduced more than 99.9% of reported polio cases. As of 2019, the World Health Organization (WHO) has recorded 143 instances of paralytic poliomyelitis globally triggered by wild polioviruses [1]. Despite the huge progress in reducing polio cases, the challenge of eradicating the last reservoirs of wild poliovirus in Pakistan and Afghanistan persists. Both countries continue to be globally endemic to poliovirus, posing a threat to the health and well-being of children [2,3]. 

To eradicate polio, Pakistan started the Polio Eradication Programme in 1994 [3]. This program is a public–private collaboration run by the federal government with the assistance of GPEI partners [4]. Each year, the Polio Eradication Programme creates National Emergency Action Plans (NEAPs) for poliovirus eradication in the country through supplementary immunization activities (SIAs) during national immunization day (NID) and subnational immunization day (SNID) campaigns. NEAP 2013 aimed at establishing oversight and ownership of the government, ensuring accountability at the district and UC levels and ensuring high-quality polio vaccination in high-risk districts and priority populations with persistent poliovirus transmission [5]. However, building on past experiences, NEAP 2021–2023 leverages the inherent strengths of Pakistan’s polio program and strategically enhances its initiatives at both the district and sub-district levels. This NEAP employs evidence-based methodologies, including risk assessments, disease modeling techniques, and the systematic categorization of districts into four levels of risk (Very High Risk, High Risk, Medium, and Low Risk), for the intervention strategy to interrupt the transmission of all polioviruses [6]. 

As a result of these efforts, type 2 wild poliovirus was eliminated from Pakistan in 2015, and the country participated in the GPEI by adopting a synchronized withdrawal of trivalent OPV (tOPV) in 2016 [7,8,9]. However, the bivalent poliovirus vaccine (bOPV) was introduced in Pakistan in 2010 and was used along with the trivalent poliovirus vaccine (tOPV) in SIAs after discounting the monovalent Sabin-strain OPV type 1 (mOPV1) and monovalent Sabin-strain OPV type 3 (mOPV3) [10,11]. Nonetheless, in 2019, when the cVDPV2 emerged in Pakistan, in addition to the monovalent Sabin-strain OPV type 2 (mOPV2), GPEI allowed the use of tOPV for epidemic response vaccination activities [7,9]. Subsequently, the 2021 NIDs and SNIDs employed bOPV, and mOPV2 or tOPV was used in areas where cVDPV2 cases were reported. 

Over the years, the number of wild poliovirus type 1 (WPV1) cases reduced from 56 in 2012 to 12 in 2018 but increased to 147 in 2019. Subsequent efforts resulted in a significant decline, with the number of cases plummeting to just two as of August 2023 [6,12]. Meanwhile, 32 cases of cVDPV were reported in 2013, but no cases were reported in 2022 [8,12]. Additionally, WHO and UNICEF estimated that, as of 2022 in Pakistan, the national vaccination among children aged 12 months for OPV3 was 85%, and for IPV1, it was 90% [13].

The oral polio vaccine (OPV) successfully eradicated wild poliovirus type 2 in 1999 and substantially reduced polio cases due to virus types 1 and 3 [14,15]. OPV has been the vaccine of choice in the fight to eradicate polio, particularly in low-income countries, because of its ease of administration, low price, and capacity to induce mucosal immunity. Nonetheless, the vaccine has certain drawbacks, including reduced immunogenicity in several tropical regions [16,17]. Furthermore, the live virus in OPV can potentially induce paralysis in those who receive the vaccine, a condition known as Vaccine-Associated Paralytic Poliomyelitis (VAPP). When in circulation, this virus can regain its neuro-virulence and result in paralysis as a Circulating Vaccine-Derived Poliovirus (cVDPV) [18,19]. In contrast, studies have demonstrated that, when compared to OPV, the inactivated poliovirus vaccine (IPV) has induced high individual immunity in countries where it has been used for routine immunization. Moreover, IPV protects against paralytic poliomyelitis without producing adverse reactions [14,20,21]. 

While it is known that immunization with IPV alone provides minimal mucosal immunity against viral shedding compared to that induced by OPV, the combined use of IPV and OPV, or the use of IPV alone in settings where the coverage with OPV is high, has been recommended [22,23]. Studies in India showed that administering a single dose of IPV substantially boosted humoral and mucosal immunity among children already primed by OPV [24,25]. However, this phenomenon has yet to be prospectively studied in Pakistan. 

We assessed the mucosal and humoral responses to poliovirus vaccines given to children who were inoculated with OPV previously. This was to measure the immunity gap among children who are at a high risk of transmitting poliovirus.

## 2. Methodology

We conducted a community-oriented, three-arm, cluster-randomized study involving healthy children under five in three high-risk polio districts of Pakistan (Karachi, Kashmore, and Bajaur) [26]. Using a computer algorithm, clusters were randomly assigned to receive standard polio program activities (bOPV) (control, arm A); enhanced interventions comprising preventive maternal and child health services, along with routine immunization, including bOPV, through health camps (arm B); or all the interventions from arm B, complemented with the administration of IPV at these health camps (arm C ~ bOPV + IPV).

We collected blood and stool samples to assess humoral and intestinal immunity among children for all three study groups. The trial was conducted between June 2013 and May 2014. The study involved the randomization of 387 clusters (131 allocated to arm A, 127 to arm B, and 129 to arm C), out of which 360 clusters remained part of the trial until its conclusion (comprising 116 in arm A, 122 in arm B, and 122 in arm C).

### 2.1. Sample Size Estimation for Humoral and Intestinal Immunity 

We calculated separate sample sizes for humoral and intestinal immunity. For humoral immunity, we assumed a baseline seroprevalence of 90%, a coverage of 90%, the immunogenicity of the bOPV arm as 50%, the immunogenicity of the bOPV + IPV arm as 90%, a power of 80%, the expected seroprevalence in the bOPV arm as 95%, the expected seroprevalence in the bOPV + IPV arm as 98%, the dropout rate as 20%, and a design effect of 2. Based on these assumptions, the sample sizes were 590 per study arm, 1770 per site, and 5310 blood samples per visit for the trial.

We assumed a baseline seroprevalence of 90%, the immunogenicity of bOPV at 80%, and the immunogenicity of the combined bOPV and IPV at 90%. Both the bOPV and bOPV + IPV arms were targeted to achieve a coverage of 90% [13,26,27,28]. With a power of 80%, we anticipated a 15% poliovirus shedding rate on day 7. We considered a dropout rate of 20% and a design effect of 2. Based on these assumptions, the sample sizes were 570 per study arm and 1710 per site, and the total sample size for the trial was 5130 stool samples per visit. Following a census of the sample size, children were selected randomly for intervention by the study team, and written consent was obtained from their parents.

### 2.2. Laboratory Methodology

Trained phlebotomists collected three milliliters (3 mL) of whole blood from each child at the three time points, i.e., before the first, second, and third rounds of immunization day supplementary immunization activities (SIAs) (Figure 1). The blood samples for the seroprevalence of polio antibodies were collected at baseline (before the first SIA), six weeks after the baseline (before the second SIA), and 18 weeks after the baseline (before the fourth SIA). Seroconversion (boosting) was assessed six weeks after the baseline (before the second SIA) and 18 weeks after the baseline (before the fourth SIA).

Blood samples underwent centrifugation, and the extracted serum was transported under cold chain conditions to the Nutrition Research Laboratory (NRL) at Aga Khan University (AKU), Karachi. The samples were stored at −20 degrees Celsius prior to their shipment to the Enterovirus Laboratory at the Centers for Disease Control and Prevention (CDC), Atlanta, Georgia, USA. Here, the levels of neutralizing antibodies were evaluated using the procedure suggested by the World Health Organization [29]. The serum underwent serial dilutions, starting at 1:8 and concluding at 1:1024. They were then incubated with 100 TCID50 of poliovirus types 1, 2, and 3 at 36 degrees Celsius for three hours before 1–2104 HEp-2 (Cincinnati) cells, which are susceptible to polioviruses, were added. Stool samples were collected before the 2nd SIA (as a baseline sample), and further samples were collected at 7 and 21 days after the 2nd SIA (Figure 1). Stool containers with ice packs were provided to the families one night before the collection by the study teams. They were transported to the NRL; AKU; and then to the Polio Reference Laboratory at the National Institutes of Health, Islamabad, Pakistan (NIH). At NIH, WHO standard procedures and guidelines were used to detect the presence of poliovirus in stool specimens [30]. All samples were transported under strict cold chain conditions.

### 2.3. Definitions of Seropositivity and Seroconversion 

Seroprevalence was defined as the proportion of subjects with titers ≥ 3 [1/dil] and calculated for each poliovirus serotype (i.e., P1, P2, and P3). Boosting was determined by a ≥4-fold increase in titers. For this research study, “immune response” encompasses boosting and seroconversion. The immune response analysis was limited to infants with an initial serological titer of ≤362 to guarantee that a 4-fold boosting response could be attainable, given that the highest titer tested was 1:1448 [26]. Intestinal immunity was represented by the composite shedding index of fecal viral titers before and 7 and 21 days after receiving a challenge dose of bOPV.

There was active surveillance for adverse events for the seven days after each vaccination and passive surveillance for up to one month post final vaccination. There was a final follow-up one month after the final vaccination. The study team actively sought safety evaluations and documented severe adverse events on designated forms. Serious adverse events—those that resulted in death, posed a threat to life, required hospitalization, or caused ongoing or significant disability—or important medical events (medical incidents that did not meet the criteria for serious adverse events but still required medical intervention) were all accounted for. Any serious adverse event or critical medical event was reported to the investigator and the data safety monitoring board without any delay.

The trial received approval from the Ethics Review Committee of Aga Khan University, Pakistan, and the National Bioethics Committee, Pakistan. Consent was obtained from the parents of the children who participated in the study. The trial has been registered on ClinicalTrials.gov under the identifier NCT01908114.

## 3. Results

Overall, this study enrolled 6429 children (3583 male and 2846 female). Of the children, 2080 were in arm A, 2171 were in arm B, and 2178 were in arm C. A total of 4915 children were aged between 24 and 59 months (Table 1). The study team collected 5982 serum samples at baseline, 4905 at visit 1, and 4564 at visit 3 (Table 2). Regional data indicate the following serum sample collections: the Karachi team collected 1793, 1565, and 1323 samples at baseline, visit 1, and visit 2, respectively; the Bajaur team collected 2005, 1532, and 1352 samples at these same intervals; and the Kashmore team collected 2184, 1806, and 1889 samples correspondingly (Appendix A). 

At baseline, serum antibody titers were higher for type 1 poliovirus than for types 2 and 3. Arm C had higher titers for all three virus types than arms A or B. Mean titers increased over time for all three virus types and in all three arms, with the largest increase in arm C. 

The key findings indicated high antibody titers against all three types, and mean titers against all three types increased in all three arms, but with the biggest increases seen in arm C (Table 2 and Figure 2). In all study groups and at all visits, P1, P2, and P3 titers were typically higher in individuals who had received a routine OPV dose than in those who had not.

To assess mucosal immunity, the study team collected 4210 stool samples at baseline (1324 in Karachi, 1371 in Bajaur, and 1515 in Kashmore), 4084 stool samples on day 7 (1143 in Karachi, 1134 in Bajaur, and 1807 in Kashmore), and 4185 stool samples on day 21 (1219 in Karachi, 1191 in Bajaur, and 1748 in Kashmore) (Table 3 and Appendix A).

The most frequently detected viruses in the stool samples at baseline were the Sabin-like virus (SL3, SL2, and SL1). The findings revealed that approximately 2–3% of stool samples were positive for SL3 and SL1 and that 1–2% were positive for SL2. 

The proportion of positive stool samples increased between baseline (day 0) and day 7 for SL1 and SL3, with the largest increase occurring in arm C (those receiving IPV). On day 21, the proportions positive for SL1 and SL3 had returned to a level similar to that seen at baseline (Figure 3). Proportions positive for SL2 were low and relatively stable. NSL1 was detected in <1% of stools at baseline (day 0), and this proportion remained low over time for all study groups. NPEV or VDPV2 was only detected in three stool samples. Viral detections of every strain were typically the highest at baseline for all age groups, but in most cases, the levels fell on day 7. Viral strains were generally detected more frequently among those who had received at least one routine dose of OPV than among those who had not.

Estimates were generated using a linear mixed model with fixed effects for the study group and visit and random effects for each child and account for cluster.

## 4. Discussion

The current study indicates that the P1 serum titer was the most elevated in Group C (IPV + OPV) at the outset, and its growth over the time was also more significant. P2 and P3 titers were generally greater in individuals who had received a regular OPV dose than in those who had not; this was observed across all study groups. Stool samples were positive for SL3 and SL1 (2–3%) and SL2 (1–2%), with the most remarkable rise from baseline occurring for SL1 and SL3 in Group C.

Our data supplement the evolving evidence, and our study is among the first studies in Pakistan that explain the significant impact of combined bOPV and IPV on intestinal and humoral immunity against poliovirus.

Several other studies in different populations have assessed the effect on humoral and intestinal immunity in children using the IPV + OPV combination. Asturias and colleagues [30] recorded 80% and 100% seroconversion in Latin American infants after three doses of bOPV combined with one or two doses of IPV and the induction of intestinal immunity against type 2 poliovirus. Likewise, a study from Chile [31] reported humoral and intestinal immunogenicity from sequential bOPV + IPV. A multicenter trial in Oman evaluated supplemental doses of IPV and reported excellent immunogenicity and increased antibody titers against poliovirus type 3. In contrast, additional doses of oral vaccines did not have these effects [32].

Complementing the well-recognized differences in OPV reactions between affluent nations and low-to-middle-income countries (LMICs) [33], several studies have been conducted in Asian contexts. An enhancement in humoral immunogenicity was noticed in Indian infants after an initial bOPV dose followed by bOPV and a singular IPV dose [34]. Bangladeshi research inferred that superior immunogenicity was achieved from sequential fractional IPV and bOPV schedules [35]. Among Sri Lankan children aged between 10 and 12 years, poliovirus was found to be excreted after challenge in 16%, 9%, and 76% of subjects from the IPV, fractional IPV (fIPV), and no IPV study arms, respectively [36]. This confirmed that a single dose of fIPV elevated mucosal immunity to a similar extent as a single full IPV dose. The study revealed that introducing IPV could address immunity deficiencies in susceptible Pakistani populations when administered alongside OPV [37]. 

The gold standard for assessing intestinal immunity is resistance to virus shedding following an oral challenge [38]. A study [39] established that the humoral response provided by the currently available IPV was greater than that provided by earlier formulations yet did not enhance intestinal immunity. The resistance to intestinal excretion depends on the challenge dose and is not absolute. In another study, 67 children who received trivalent OPV and were followed up for ten years documented declining serum antibody titers indicating decreasing resistance to intestinal excretion [40], reflecting that intestinal immunity is temporary [41,42]. The research findings indicate a mild correlation between pre-challenge antibody levels after vaccination with either IPV or bOPV + IPV and variations in gut immunity, which is insufficient to forecast intestinal immunity against polio type 2 [43].

Existing research indicates a constrained role of IPV in enhancing mucosal antibodies and curbing poliovirus shedding among those not previously exposed to live viruses [23,30,31,44,45,46,47,48]. Despite receiving zero, two, or three prior IPV doses in Cuba, over 90% of infants shed some form of poliovirus following the tOPV challenge [48]. The ability of the inactivated polio vaccine (IPV) to trigger a primary mucosal immune response, capable of inhibiting the replication of live polio and thereby controlling the transmission of poliovirus, remains uncertain. A Phase 2 clinical trial with infants from Panama found that IPV-induced serum neutralization does not significantly enhance intestinal mucosal immunity or restrict viral shedding following a monovalent type 2 OPV challenge [49]. Still, it might reduce the quantity and duration of shedding [50,51].

Studies focusing on poliovirus-specific immunoglobulin A have revealed that exposure to live poliovirus through OPV or the environment is essential in inducing mucosal responses to IPV [46,47]. A systematic review has pointed out that, when IPV is delivered without OPV, it statistically significantly fails to reduce the odds of fecal shedding following a challenge dose with live attenuated polioviruses [23]. Similarly, a pair of clinical studies from India have indicated that a singular IPV dose administered to children previously immunized with OPV considerably amplified their defense against poliovirus shedding following a subsequent OPV challenge [24,25]. Macklin and colleagues also observed a boost in mucosal protection induced by IPV in OPV-primed individuals, establishing that IPV works in a serotype-specific model [52]. Moreover, a meta-analysis showed the post-vaccination dependence of shedding on several vaccine doses and pre-challenge titers [53].

However, the literature also revealed that three doses of IPV without bOPV induced more significant quantities of virus shedding than fewer IPV doses [54]. Evidence from bOPV/IPV-integrated trials confirmed a potentially substantial role of IPV in the stimulation of mucosal immunity. In Latin America, a group of infants were administered bOPV at 6, 10, and 14 weeks, along with an extra IPV dose at 14 weeks. These infants exhibited higher type 2-specific stool neutralization during the mOPV2 challenge and lower viral shedding than their counterparts who received bOPV only [30,45]. It is noteworthy that, since IPV offers a more limited mucosal immune response than OPV [22,31], there remains a possibility for poliovirus circulation in populations immunized exclusively with IPV without causing poliomyelitis, as was reported in Israel [55]. 

Furthermore, the novel approach of incorporating *E. coli* labile toxin with double attenuating mutations (dmLT) to improve the immunologic responses to IPV at mucosal sites was examined. A quadrupling of serotype-specific neutralizing antibody (SNA) titers was observed for all three serotypes in 84% of subjects receiving the fractional-dose inactivated polio vaccine (fIPV-dmLT) compared to just 50% of participants who received IPV alone. This finding underscores the advantage of using fIPV-dmLT over IPV by itself [56].

## 5. Conclusions

For populations with high rates of OPV failure, an IPV booster inoculated following a minimum of two OPV doses could potentially reduce immunity gaps. However, IPV alone does not provide enough intestinal immunity to prevent the infection and spread of live poliovirus among populations without prior exposure. Future research should investigate the connection between mucosal immunity and serum antibodies to understand virus spread risks in IPV-immunized populations, helping to devise strategies to control poliovirus shedding and prevent outbreaks.

## Figures and Tables

**Figure 1 vaccines-11-01444-f001:**
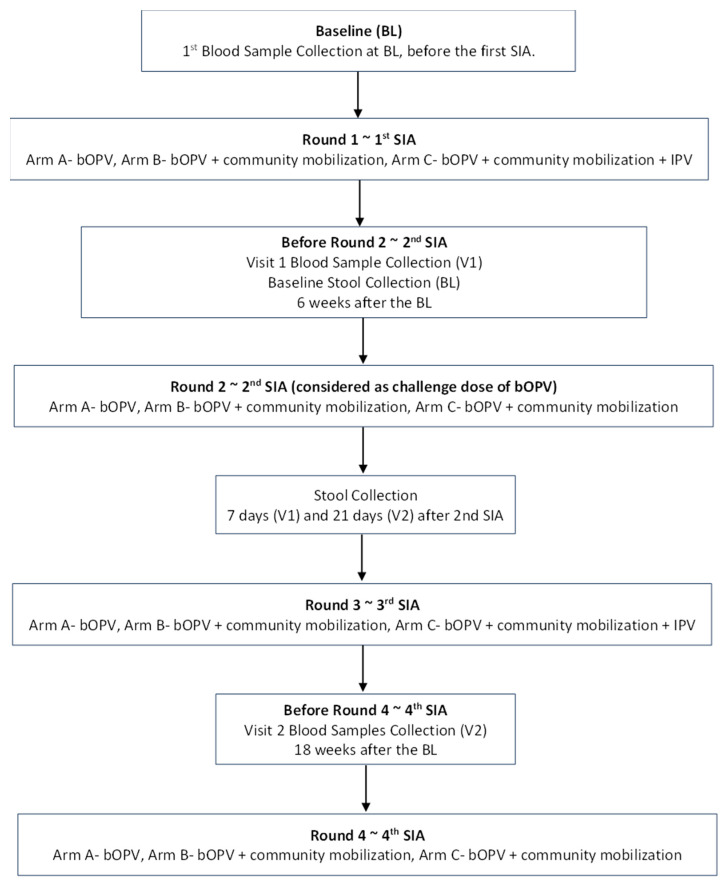
Flow Diagram of Field Activities.

**Figure 2 vaccines-11-01444-f002:**
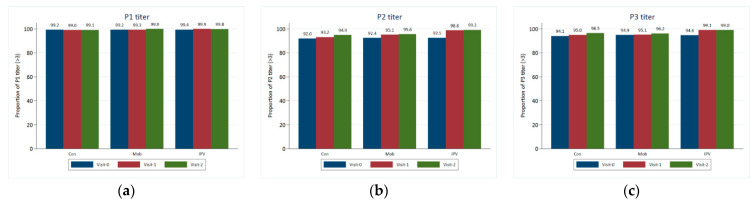
Crude proportion of children with P1, P2, and P3 titers >= 3.00 by study group and visit. (**a**) Proportion of children with a P1 titer ≥ 3.00 by study group and visit; (**b**) proportion of children with a P2 titer ≥ 3.00 by study group and visit; (**c**) proportion of children with a P3 titer ≥ 3.00 by study group and visit.

**Figure 3 vaccines-11-01444-f003:**
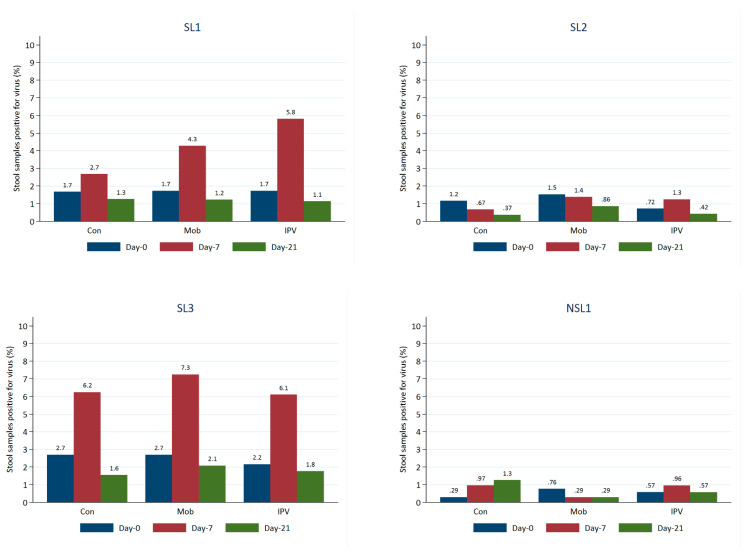
Proportion of stool samples positive for viruses over time by virus and study group.

**Table 1 vaccines-11-01444-t001:** Demographic characteristics of children by study group.

	Overall	A	B	C
	N = 6429	N = 2080	N = 2171	N = 2178
**Gender**				
Male	3583 (55.7)	1164 (56.0)	1202 (55.4)	1217 (55.9)
Female	2846 (44.3)	916 (44.0)	969 (44.6)	961 (44.1)
**Age group**				
0–23 months	1514 (23.5)	496 (23.8)	503 (23.2)	515 (23.6)
24–59 months	4915 (76.5)	1584 (76.2)	1668 (76.8)	1663 (76.4)
Age in months, mean ± SD	33.3 ± 14.7	32.9 ± 14.7	33.7 ± 14.6	33.3 ± 14.8

SD: standard deviation.

**Table 2 vaccines-11-01444-t002:** Estimated mean P1, P2, and P3 titers by study group and visit.

	P1 Titer (95% CI)			P2 Titer (95% CI)			P3 Titer (95% CI)		
Group	A	B	C	A	B	C	A	B	C
**Visit**									
**Baseline** **N = 5982**	8.48 ^†₸^	8.55 *^₸^	8.68 †*^‖£^	6.74 ^†₸^	6.71 *^₸^	6.89 ^†^*^₸^	6.90 ^†₸^	6.97 *^₸^	7.9 ^†^*^₸^
	(8.41, 8.55)	(8.48, 8.61)	(8.61, 8.74)	(6.65, 6.83)	(6.62, 6.80)	(6.80, 6.97)	(6.81, 6.99)	(6.88, 7.06)	(7.01, 7.18)
**Visit 1** **N = 4905**	8.56 ^†₸^	8.66 *^₸^	9.91 ^†^*^‖^	7.07 ^‡₸^	7.21 ^‡₸^	9.25 ^‡₸^	7.08 ^†₸^	7.18 *^₸^	9.55 ^†^*^₸^
	(8.49, 8.64)	(8.59, 8.73)	(9.83, 9.98)	(6.98, 7.17)	(7.12, 7.31)	(9.15, 9.35)	(6.98, 7.18)	(7.08, 7.27)	(9.46, 9.65)
**Visit 2** **N = 4564**	8.78 ^‡₸^	8.89 ^‡₸^	9.93 ^‡£^	7.25 ^†₸^	7.32 *^₸^	9.38 ^†^*^₸^	7.6 ^†₸^	7.70 *^₸^	9.72 ^†^*^₸^
	(8.70, 8.85)	(8.82, 8.96)	(9.86, 10.01)	(7.15, 7.36)	(7.23, 7.42)	(9.28, 9.48)	(7.50, 7.70)	(7.60, 7.79)	(9.62, 9.82)

Estimates were generated using a linear mixed model with fixed effects for the study group and visit and random effects for each child and account for cluster. **Groups:** A = control, B = Community Mobilization, C = Community Mobilization + IPV. **Abbreviations:** CI: Confidence Interval, P1: poliovirus type 1, P2: poliovirus type 2, P3; poliovirus type 3. † *p*-value < 0.05 for Groups A and C. * *p*-value < 0.05 for Groups B and C. ‡ *p*-value < 0.05 for A and B, B and C, A and C. ‖ *p*-value < 0.05 for baseline and Visit-1. £ *p*-value < 0.05 for baseline and Visit-2. ^₸^
*p*-value < 0.05 for baseline and Visit-1, baseline and Visit-2, Visit-1 and Visit-2.

**Table 3 vaccines-11-01444-t003:** Number of children with stool samples positive for viruses by virus and study group.

		Group A		Group B		Group C	
Virus	Visit	N	*n* (%)	N	*n* (%)	N	*n* (%)
SL1 (Sabin-like virus 1)	Day-0	1373	23 (1.7)	1445	25 (1.7)	1392	24 (1.7)
	Day-7	1345	36 (2.7) †**	1379	59 (4.3) **	1360	79 (5.8) †
	Day-21	1353	17 (1.3)	1391	17 (1.2)	1414	16 (1.1)
SL2 (Sabin-like virus 2)	Day-0	1373	16 (1.2)	1445	22 (1.5)	1392	10 (0.7)
	Day-7	1345	9 (0.7)	1379	19 (1.4)	1360	17 (1.3)
	Day-21	1353	5 (0.4)	1391	12 (0.9)	1414	6 (0.4)
SL3 (Sabin-like virus 3)	Day-0	1373	37 (2.7)	1445	39 (2.7)	1392	30 (2.2)
	Day-7	1345	84 (6.2)	1379	100 (7.3)	1360	83 (6.1)
	Day-21	1353	21 (1.6)	1391	29 (2.1)	1414	25 (1.8)
NSL1 (non-Sabin-like virus)	Day-0	1373	4 (0.3)	1445	11 (0.8)	1392	8 (0.6)
	Day-7	1345	13 (1.0)	1379	4 (0.3)	1360	13 (1.0)
	Day-21	1353	17 (1.3) †**	1391	4 (0.3) **	1414	8 (0.6) †
NPEVs (non-polio enteroviruses)	Day-0	1373	0	1445	0	1392	1 (0.1)
	Day-7	1345	0	1379	0	1360	0
	Day-21	1353	0	1391	1 (0.1)	1414	0
VDPV2 (vaccine-derived poliovirus 2)	Day-0	1373	0	1445	1 (0.1)	1392	0
	Day-7	1345	0	1379	0	1360	0
	Day-21	1353	0	1391	0	1414	0

**Groups:** A = control, B = Community Mobilization, C = Community Mobilization + IPV. **Abbreviations:** N = total sample size, *n* = number with a positive stool sample., † *p*-value < 0.05 for Groups A and C. ** *p*-value < 0.05 for Groups A and B.

## Data Availability

The data presented in this study are available on request from the corresponding author. The data are not publicly available due to privacy and ethical concerns.

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
