# Peer review of "Does IPV Boost Intestinal Immunity among Children under Five Years of Age? An Experience from Pakistan"

_vaccines, 2023, doi:10.3390/vaccines11091444_

Round 1
Reviewer 1 Report
The article is interesting but there are some points to be presented :
- The information about polio eradication status in the world as: Pakistan, Afghanistan are the only two countries left in the world where the poliovirus continues to threaten the health and well-being of children. https://www.endpolio.com.pk/polioin-pakistan
- A short history of polio immunization in Pakistan and the role of the National Eradication Strategic Plan, a comparison between the plan from 2013 to 2021 -2023. It is unclear the year of introduction of bOPV (types 1 and 3) in Pakistan. The Strategic Goal for 2021-2023 is to permanently interrupt all poliovirus transmission in Pakistan by the end of 2023. https://polioeradication.org/wp-content/uploads/2022/02/NEAP-2021-2023.pdf
- The sentence from rows 49-50 that IPV protects against paralytic poliomyelitis, especially in low-income countries, can create confusion. The main advantage of IPV use is that vaccination does not produce adverse reactions as emergency of VDPVs and the VAPP cases, and one of the disadvantage is that it is an expensive vaccine.
- The sentences about “assuming a baseline seroprevalence of 90%, the immunogenicity of bOPV at 80%, and the combined bOPV and IPV at 90%. and “anticipated” a 15% of PV shedding rate must be scientifically proved (rows 80-82).
The World Health Organization (WHO) and UNICEF estimated Pakistan’s 2021 national polio vaccination coverage (3 doses of oral poliovirus vaccine [OPV] and 1 dose of inactivated poliovirus vaccine by age 12 months) at 83%.After the declaration of eradication of WPV type 2 in 2015, Pakistan joined other countries in GPEI in implementing a synchronized withdrawal of trivalent OPV (tOPV; containing Sabin-strain types 1, 2, and 3) in 2016 as part of containment efforts for all type 2 polioviruses. However, with the emergence of cVDPV2 in Pakistan in 2019, GPEI authorized the use of tOPV along with the recommended monovalent Sabin-strain OPV type 2 (mOPV2) for outbreak response vaccination activities. During 2021, 4 national immunization days (NIDs) and 2 subnational immunization days (SNIDs) directed at children aged <5 years were conducted using bivalent OPV (bOPV; containing Sabin-strain types 1 and 3) and, in areas with cVDPV2 transmission, either mOPV2 or tOPV. Mbaeyi C, Baig S, Safdar MR, et al. Progress Toward Poliomyelitis Eradication — Pakistan, January 2021–July 2022. MMWR Morb Mortal Wkly Rep 2022;71:1313–1318.DOI: http://dx.doi.org/10.15585/mmwr.mm7142a1
- The Pakistan’s progress toward polio eradication during January 2021–July 2022 were presented in reports:
Mbaeyi C, Baig S, Khan Z, et al. Progress toward poliomyelitis eradication—Pakistan, January 2020–July 2021. MMWR Morb Mortal Wkly Rep 2021;70:1359–64. https://doi.org/10.15585/mmwr.mm7039a1PMID:34591827
Hsu CH, Rehman MS, Bullard K, et al. Progress toward poliomyelitis eradication—Pakistan, January 2019−September 2020. MMWR Morb Mortal Wkly Rep 2020;69:1748–52. https://doi.org/10.15585/mmwr.mm6946a5PMID:33211676
In conclusion the information must be brought up to date, as well as the bibliography and also the results obtained in the study carried out in the period 2013-2014 must be compared with those existing in 2022-2023 at the global level.
Author Response
Comments from Reviewer1:
The article is interesting but there are some points to be presented:
Comment 1: The information about polio eradication status in the world as: Pakistan, and Afghanistan are the only two countries left in the world where the poliovirus continues to threaten the health and well-being of children. https://www.endpolio.com.pk/polioin-pakistan
Response: Thank you for the valuable input. We added the revised sentence “…the challenge of eradicating the last reservoirs of wild poliovirus in Pakistan and Afghanistan persists. Both countries continue to be globally endemic to poliovirus, posing a threat to the health and well-being of children [2,3]”, and the following reference.
- Pakistan Polio Eradication Program, Polio in Pakistan, https://www.endpolio.com.pk/polioin-pakistan [accessed August 11, 2023].
Comment 2: A short history of polio immunization in Pakistan and the role of the National Eradication Strategic Plan, a comparison between the plan from 2013 to 2021 -2023. It is unclear the year of introduction of bOPV (types 1 and 3) in Pakistan. The Strategic Goal for 2021-2023 is to permanently interrupt all poliovirus transmission in Pakistan by the end of 2023. https://polioeradication.org/wp-content/uploads/2022/02/NEAP-2021-2023.pdf
Response: We thank the reviewer for the insightful comment. Based on the suggestion, we added the following paragraph along with the listed references.
“The polio eradication efforts in Pakistan began in 1994, through the launch of the Pakistan Polio Eradication Programme [3]. The Programme is a public–private partnership led by the federal government and supported by GPEI partners including WHO, UNICEF, Bill and Melinda Gates Foundation (BMFG), and the Centers for Disease Control and Prevention (CDC) [4]. The program develops annual National Emergency Action Plans (NEAP) for poliovirus eradication in the country through Supplementary Immunization activities (SIAs) during national immunization days (NIDs) and subnational immunization days (SNIDs) campaigns. NEAP 2013 aimed at establishing oversight and ownership of the government, ensuring accountability at the district and UC levels and ensuring high-quality polio vaccination in high-risk districts and priority populations with persistent poliovirus transmission [5]. Whereas, building on past experiences, the NEAP 2021-2023 leverages the inherent strengths of Pakistan's polio program and strategically enhance its initiatives at both the district and sub-district levels. This NEAP employs evidence-based methodologies, including risk assessments and disease modelling techniques, and systematically categorizes districts into four levels of risk: Very High-Risk, High-Risk, Medium, and Low-Risk for intervention strategy to interrupt transmission of all polioviruses [6].
As a result of these efforts, the WPV type 2 was eradicated from Pakistan in 2015, and the country joined other countries in GPEI in implementing a synchronized withdrawal of trivalent OPV (tOPV) in 2016 as part of containment efforts for all type 2 polioviruses [7,8,9]. Although, the bivalent poliovirus vaccine (bOPV) was introduced in Pakistan in 2010 and used along with trivalent poliovirus vaccine (tOPV) in SIAs after discounting the monovalent Sabin-strain OPV type 1 (mOPV1) and monovalent Sabin-strain OPV type 3 (mOPV3) [10,11]. However, with the emergence of cVDPV2 in Pakistan in 2019, GPEI authorized the use of tOPV along with the recommended monovalent Sabin-strain OPV type 2 (mOPV2) for outbreak response vaccination activities [7,9]. Subsequently, the 2021 NIDs and SNIDs employed bOPV, and in areas with cVDPV2 transmission, either mOPV2 or tOPV was used.”
- Pakistan Polio Eradication Program, Polio in Pakistan, https://www.endpolio.com.pk/polioin-pakistan [accessed August 11, 2023].
- Pakistan Polio Eradication Programme. Partners and donors, https://www.endpolio.com.pk/polioin-pakistan/partners-and-donors. [accessed August 11, 2023].
- Government of Islamic Republic of Pakistan. “National Emergency Action Plan For Polio Eradication 2013.” https://www.endpolio.com.pk/images/reports/National-Emergency-Action-Plan-2013-(NEAP-2013).pdf [Accessed August 11, 2023].
- National Emergency Operation Centre. Pakistan Polio Eradication Initiative National Emergency Action Plan 2021-2023. https://polioeradication.org/wp-content/uploads/2022/02/NEAP-2021-2023.pdf. [August 11, 2023].
- Hsu CH, Rehman MS, Bullard K, et al. Progress toward poliomyelitis eradication—Pakistan, January 2019−September 2020. MMWR Morb Mortal Wkly Rep 2020;69:1748–52. https://doi.org/10.15585/mmwr.mm6946a5 PMID:33211676
- Mbaeyi C, Baig S, Khan Z, et al. Progress toward poliomyelitis eradication—Pakistan, January 2020–July 2021. MMWR Morb Mortal Wkly Rep 2021;70:135964. https://doi.org/10.15585/mmwr.mm7039a1 PMID:34591827
x9. Mbaeyi C, Baig S, Safdar MR, et al. Progress Toward Poliomyelitis Eradication — Pakistan, January 2021–July 2022. MMWR Morb Mortal Wkly Rep 2022;71:1313–1318.DOI: http://dx.doi.org/10.15585/mmwr.mm7142a1
- Alexander JP Jr, Zubair M, Khan M, Abid N, Durry E. Progress and peril: poliomyelitis eradication efforts in Pakistan, 1994-2013. J Infect Dis. 2014 Nov 1;210 Suppl 1(Suppl 1):S152-61. doi: 10.1093/infdis/jiu450. PMID: 25316830; PMCID: PMC6261452.
- Centre for Disease Control and Prevention (CDC). Progress toward poliomyelitis eradication--Afghanistan and Pakistan, January 2010-September 2011. MMWR Morb Mortal Wkly Rep. 2011 Nov 11;60(44):1523-7. PMID: 22071591.
Comment 3: The sentence from rows 49-50 that IPV protects against paralytic poliomyelitis, especially in low-income countries, can create confusion. The main advantage of IPV use is that vaccination does not produce adverse reactions as emergency of VDPVs and VAPP cases, and one of the disadvantages is that it is an expensive vaccine.
Response: Thank you for a thorough review. The sentence has been revised to address the confusion as below:
“… when compared to OPV, the inactivated poliovirus vaccine (IPV) has induced a high individual immunity in countries where it has been used for routine immunization. Moreover, IPV protects against paralytic poliomyelitis without producing adverse reactions [14,20,21]”.
Comment 4: The sentences about “assuming a baseline seroprevalence of 90%, the immunogenicity of bOPV at 80%, and the combined bOPV and IPV at 90%. and “anticipated” a 15% of PV shedding rate must be scientifically proved (rows 80-82).
Response: Thank you for highlighting this crucial point. Based on administrative data indicating a 95% coverage 2013 and 90% in 2014 for the third dose of OPV in Pakistan, we considered a seroprevalence of 90%. Regarding the immunogenicity of bOPV at 80%, as well as the combined bOPV and IPV at 90%, and the anticipated 15% PV shedding, our assumptions draw from existing literature. These references and assumptions have been appropriately documented in the manuscript.
- Immunization Pakistan 2023 Country Profile. https://www.who.int/publications/m/item/immunization-pakistan-2023-country-profile. [accessed August 12, 2023].
- Habib MA, Soofi S, Cousens S, Anwar S, ul Haque N, Ahmed I, Ali N, Tahir R, Bhutta ZA. Community engagement and integrated health and polio immunisation campaigns in conflict-affected areas of Pakistan: a cluster randomised controlled trial. The Lancet Global Health. 2017 Jun 1;5(6): e593-603.
- Sutter RW, John TJ, Jain H, et al. Immunogenicity of bivalent types 1 and 3 oral poliovirus vaccine: a randomised, double-blind, controlled trial. The Lancet. 2010;376(9753):1682-1688. doi:1016/S0140-6736(10)61230-5
- Saleem AF, Mach O, Quadri F, et al. Immunogenicity of poliovirus vaccines in chronically malnourished infants: A randomized controlled trial in Pakistan. Vaccine. 2015;33(24):2757-2763. doi:1016/j.vaccine.2015.04.055
Comment 5: The World Health Organization (WHO) and UNICEF estimated Pakistan’s 2021 national polio vaccination coverage (3 doses of oral poliovirus vaccine [OPV] and 1 dose of inactivated poliovirus vaccine by age 12 months) at 83%.After the declaration of eradication of WPV type 2 in 2015, Pakistan joined other countries in GPEI in implementing a synchronized withdrawal of trivalent OPV (tOPV; containing Sabin-strain types 1, 2, and 3) in 2016 as part of containment efforts for all type 2 polioviruses. However, with the emergence of cVDPV2 in Pakistan in 2019, GPEI authorized the use of tOPV along with the recommended monovalent Sabin-strain OPV type 2 (mOPV2) for outbreak response vaccination activities. During 2021, 4 national immunization days (NIDs) and 2 subnational immunization days (SNIDs) directed at children aged <5 years were conducted using bivalent OPV (bOPV; containing Sabin-strain types 1 and 3) and, in areas with cVDPV2 transmission, either mOPV2 or tOPV.
Mbaeyi C, Baig S, Safdar MR, et al. Progress Toward Poliomyelitis Eradication — Pakistan, January 2021–July 2022. MMWR Morb Mortal Wkly Rep 2022;71:1313–1318. DOI: http://dx.doi.org/10.15585/mmwr.mm7142a1.
- The Pakistan’s progress toward polio eradication during January 2021–July 2022 were presented in reports:
Mbaeyi C, Baig S, Khan Z, et al. Progress toward poliomyelitis eradication—Pakistan, January 2020–July 2021. MMWR Morb Mortal Wkly Rep 2021;70:135964. https://doi.org/10.15585/mmwr.mm7039a1 PMID:34591827
Hsu CH, Rehman MS, Bullard K, et al. Progress toward poliomyelitis eradication—Pakistan, January 2019−September 2020. MMWR Morb Mortal Wkly Rep 2020;69:1748–52. https://doi.org/10.15585/mmwr.mm6946a5 PMID:33211676
In conclusion, the information must be brought up to date, as well as the bibliography and also the results obtained in the study carried out in the period 2013-2014 must be compared with those existing in 2022-2023 at the global level.
Response: We thank the reviewer for the valuable input. Part of this comment and suggested references are addressed in response to comment 2. And to update the polio epidemiology, we have added the following paragraph along with relevant references.
“Over the years, the number of wild poliovirus type-1 (WPV1) cases reduced from 56 in 2012 to 12 in 2018 but increased to 147 in 2019. Subsequent efforts resulted in a significant decline, with cases plummeting to just 2 as of August 2023 [6,12]. Meanwhile, 32 cases of cVDPV were reported in 2013 but no cases were reported in 2022 [8,12]. Additionally, WHO and UNICEF estimated that in 2022, national vaccination for 3 doses of oral poliovirus vaccine [OPV] and 1 dose of IPV by age 12 months in Pakistan was 85% and 90% respectively [13].”
- National Emergency Operation Centre. Pakistan Polio Eradication Initiative National Emergency Action Plan 2021-2023. https://polioeradication.org/wp-content/uploads/2022/02/NEAP-2021-2023.pdf. [August 11, 2023].
- Pakistan Polio Eradication Program; Polio Cases in Provinces.” https://www.endpolio.com.pk/polioin-pakistan/polio-cases-in-provinces. [accessed August 12, 2023].
- Immunization Pakistan 2023 Country Profile. https://www.who.int/publications/m/item/immunization-pakistan-2023-country-profile. [accessed August 12, 2023].
Reviewer 2 Report
This paper provides valuable information for the global polio eradication program. However, some points about the figure and the table need to be improved and clarified before being accepted.
Minor
1.line 13: The font of "Oral polio vaccine" is different from the other sentences.
2.line22: I think that the word ")" of "bOPV)" is not necessary.
3.line27: What does ”P2 and P3” mean? Poliovirus type 2 and type 3?
4.Figure1: Does it mean anything that the arrows are out of position?
5.Table1: What does ”P1, P2 and P3” mean? Poliovirus type 1, 2 and type 3? The font of p-value was different.
6. Figure2, 3 and 4: Is "Group A, B and C on figure2" same "Group Con, Mob and IPV on figure 3 and 4"?
Author Response
Comments from Reviewer 2:
This paper provides valuable information for the global polio eradication program. However, some points about the figure and the table need to be improved and clarified before being accepted.
Comment 1: Line 13: The font of "Oral polio vaccine" is different from the other sentences.
Response: Thank you for pointing this out, it is corrected in the revised manuscript.
Comment 2: Line22: I think that the word ")" of "bOPV)" is not necessary.
Response: Yes, it was redundant, it is removed in the revised version of the manuscript.
Comment 3: line27: What does “P2 and P3” mean? Poliovirus type 2 and type 3?
Response: Thank you for highlighting this. We have included the following clarification: “Poliovirus type 2 (P2) and Poliovirus type 3 (P3)”.
Comment 4: Figure1: Does it mean anything that the arrows are out of position?
Response: Thank you, the positioning of the arrows in the figure has been rectified.
Comment 5: Table1: What does “P1, P2 and P3” mean? Poliovirus type 1, 2 and type 3? The font of p-value was different.
Response: Thank you for highlighting this, the abbreviations are explained in the revised manuscript as provided below. Also, the font size of the p-value is corrected.
“P1: Poliovirus type 1, P2: Poliovirus type 2, P3; Poliovirus type 3”.
Comment 6: Figure2, 3 and 4: Is "Group A, B and C on figure2" same as "Group Con, Mob and IPV on figure 3 and 4"?
Response: We sincerely apologize for the confusion created. To clarify, yes group A, B, and C in figure 2 correspond to group Con, Mob and IPV respectively, as depicted in Figures 3 and 4.
Reviewer 3 Report
The paper by Atif Habib et al address an important topic on polio immunization. However, I have significant comments.
1. The Introduction is OK. A short description of the standard polio program would be helpful
2. The Methodology section should be improved by subheadings for clarity. A short description on the number of girls vs boys and age at inclusion in the different arms should be added. It is unclear since the title only says “among children under 5 years of age”. Had the children received OPV before entering this study and in what time frame?
a. Statistics should be a separate heading. The section regarding calculation s for the intestinal immunity should be corrected p2line 85 for blood samples . I assume it should be stool samples.
b. Sampling of patients should be a separate heading where blood as well as stool sampling is described. Please explain p2 line 87, was there a selected sampling of children before the study?
c. Handling of the samples should be one section for clarity.
d. Definitions of seropositivity and seroconversion requires a specific subheading so that readers easily can understand the work.
3. The result section should be clearly marked. Table 1 is OK but please indicate N= XXXx for the numbers given under the different timepoints. I fail to see how table 1 and Fig 2 is different-please explain or choose one of these. Figure 3 needs an explanation, why do you show titers >=3 as stated in the figure legend. In the methodology section, seropositivity was defined as >8.
P5 line 16 says “ In all study groups and at all visits…..titers were higher in individuals who had received a routine OPV dose compared to those who had not”. Where is that previously described+ how many children missed the routine OPV dose? Needs clarification!!
P6 l18 last sentence also needs clarificiation in terms of number of children who did not receive a routine dose of OPV.
The discussion should be more focused on the results generated for this particular study.
Please check language for stringency.
Author Response
Comments from reviewer 3:
The paper by Atif Habib et al address an important topic on polio immunization. However, I have significant comments.
Comment 1: The Introduction is OK. A short description of the standard polio program would be helpful.
Response: Thank you for the feedback, we have added the following paragraph about the polio program in the introduction:
“The polio eradication efforts in Pakistan began in 1994, through the launch of the Pakistan Polio Eradication Programme [3]. The Programme is a public–private partnership led by the federal government and supported by GPEI partners including WHO, UNICEF, Bill and Melinda Gates Foundation (BMFG), and the Centers for Disease Control and Prevention (CDC) [4]. The program develops annual National Emergency Action Plans (NEAP) for poliovirus eradication in the country through Supplementary Immunization activities (SIAs) during national immunization days (NIDs) and subnational immunization days (SNIDs) campaigns.”
Comment 2: The Methodology section should be improved by subheadings for clarity. A short description on the number of girls vs boys and age at inclusion in the different arms should be added. It is unclear since the title only says “among children under 5 years of age”. Had the children received OPV before entering this study and in what time frame?
- Statistics should be a separate heading. The section regarding calculations for intestinal immunity should be corrected p2line 85 for blood samples. I assume it should be stool samples.
- Sampling of patients should be a separate heading where blood, as well as stool sampling, is described. Please explain p2 line 87, was there a selected sampling of children before the study?
- Handling of the samples should be one section for clarity.
- Definitions of seropositivity and seroconversion requires a specific subheading so that readers easily can understand the work.
Response: Thank you for the valuable feedback. We have added Table 1 as a descriptive table providing the gender and age-wise number of children included in the study in each of the three arms. This trial study assigned the children to the three arms to assess the immunity gap between June 2013 and May 2014; 1) Control, arm A: received standard polio program activities (bOPV). 2) arm B: enhanced interventions comprising preventive maternal and child health services along with routine immunization, including bOPV, via health camps. 3) arm C: the interventions from arm B, complemented with the administration of IPV at these health camps (bOPV + IPV).
- The following subheadings are added to the methodology section: “Sample sizes estimation for humoral and intestinal immunity, Laboratory methodology, Definitions of seropositivity and seroconversion.”
And the point regarding the calculations for intestinal immunity at p2 line 85 is corrected as “stool samples”.
- For p2 line 87; No, there was no selected sample before the study, rather the sample size was calculated separately for humoral and intestinal immunity, and children were randomly selected for the intervention. We have revised the sentence as;
“Following a census on the sample size, children were selected randomly for intervention by the study team…”
- Thank you, a subheading “Laboratory methodology” is added in the methodology section.
- The revised manuscript contains a sub-heading “Definitions of seropositivity and seroconversion.”
Comment 3: The result section should be clearly marked. Table 1 is OK but please indicate N= XXXx for the numbers given under the different timepoints. I fail to see how Table 1 and Fig 2 is different-please explain or choose one of these. Figure 3 needs an explanation, why do you show titers >=3 as stated in the figure legend. In the methodology section, seropositivity was defined as >8.
P5 line 16 says “ In all study groups and at all visits…..titers were higher in individuals who had received a routine OPV dose compared to those who had not”. Where is that previously described+ how many children missed the routine OPV dose? Needs clarification!!
P6 l18 last sentence also needs clarification in terms of number of children who did not receive a routine dose of OPV. The discussion should be more focused on the results generated for this particular study.
Response: Thank you for a thorough review. The result section has clearly marked as a separate heading. As per the suggestion “N” is added for the numbers in Table 1 under the different time points. We do agree that Table 1 and Fig 2 are the same, and we have removed Fig 2 in the revised manuscript. We revised the definition of seroprevalence in the methodology to reflect the graphs in the figure 3.
Reviewer 4 Report
14-15&48-49
"the immunity induced by inactivated poliovirus vaccine is better than that induced by OPV" - What you mean? Please, reformulate or explain.
22&64
"bOPV" - please, decipher this term and explain why did you use bivalent vaccine rather than trivalent before the switch in 2016. I am sure that the article should have an explanation about the difference between bOPV and tOPV (OPV) and the comments about the global switch from OPV to bOPV.
23-24&66
Why do you think that the immune response of children in arm A and arm B should be different? What was "the preventive maternal and child health service" exactly?
91
SIA - please, decipher this term.
96
It is advisable to include the timing of various manipulations in the diagram (Fig.1)
127-131
Could you list the "serious adverse events or important medical events", at least in supplementary?
Author Response
Comments from Reviewer 4:
Comment 1: 14-15&48-49 "the immunity induced by inactivated poliovirus vaccine is better than that induced by OPV" - What you mean? Please, reformulate or explain.
Response: Thank you for a thorough review, for lines 14-15 the sentence is revised as “However, the immunity induced by inactivated poliovirus vaccine (IPV) is better than that induced by OPV, as IPV protects against paralytic poliomyelitis without producing adverse reactions”.
And for lines 48-49 the sentence is revised as “In contrast, studies have demonstrated that, when compared to OPV, the inactivated poliovirus vaccine (IPV) has induced a high individual immunity in countries where it has been used for routine immunization. Moreover, IPV protects against paralytic poliomyelitis without producing adverse reactions.”
Comment 2: 22&64 "bOPV" - please, decipher this term and explain why did you use bivalent vaccine rather than trivalent before the switch in 2016. I am sure that the article should have an explanation about the difference between bOPV and tOPV (OPV) and the comments about the global switch from OPV to bOPV.
Response: Thank you for pointing out this key aspect. In this study, we employed the bivalent oral poliovirus vaccine (bOPV) before the global switch from trivalent oral poliovirus vaccine (tOPV) to assess its impact on intestinal and humoral immunity against poliovirus in Pakistan. The choice to use bOPV instead of tOPV before the 2016 switch was due to bOPV containing two serotypes of poliovirus (types 1 and 3), excluding type 2. This was aimed to eliminate the risk of vaccine-derived polio cases associated with type 2. The article covers the distinctions between bOPV and tOPV and discusses the rationale behind the global switch to bOPV.
Also, the explanation of “bOPV” is added in line 24.
Top of Form
Comment 3: 23-24&66; Why do you think that the immune response of children in arm A and arm B should be different? What was "the preventive maternal and child health service" exactly?
Response: Once again we thank the reviewer for the insightful comment. The preventive maternal and child health services included nutritional screening services for women and children, ante-natal, post-natal care services and Tetanus vaccination for women and essential immunization for children. We believe the immune responses of the children in arm A and arm B should differ. This is because children in Arm A only received bOPV through the routine polio program, whereas children in Arm B received polio vaccinations during the health camp, in addition to bOPV through the routine polio program.
Comment 4: 91; SIA - please, decipher this term.
Response: Thank you for highlighting this. We added supplementary immunization activities (SIA) to the revised text.
Comment 5: 96; It is advisable to include the timing of various manipulations in the diagram (Fig.1).
Response: Thank you for providing valuable input. The figure now includes the timing of sample collection at various intervals.
Comment 6: 127-131; Could you list the "serious adverse events or important medical events", at least in supplementary?
Response: Thank you for the comment. We didn’t encounter any adverse or important medical events in our study. However, in compliance with ethical research requirements, we conducted safety evaluations of the participants and established a robust mechanism to document any severe adverse events should they occur.
Round 2
Reviewer 1 Report
The authors entered the requested information in the article.
Author Response
Thank you for showing satisfaction with the responses.
Reviewer 3 Report
The authors have adressed my concerns and the manuscript is improved. I have no further comments
Author Response

(The authors gave the same response as above.)
